# Autophagy—A Story of Bacteria Interfering with the Host Cell Degradation Machinery

**DOI:** 10.3390/pathogens10020110

**Published:** 2021-01-22

**Authors:** Anna K. Riebisch, Sabrina Mühlen, Yan Yan Beer, Ingo Schmitz

**Affiliations:** 1Department of Molecular Immunology, Ruhr-University Bochum, 44801 Bochum, Germany; anna.riebisch@ruhr-uni-bochum.de (A.K.R.); sabrina.muehlen@ruhr-uni-bochum.de (S.M.); 2Institute for Infectiology, University of Münster, 48149 Münster, Germany; 3German Center for Infection Research (DZIF), Associated Site University of Münster, 48149 Münster, Germany; 4Systems-Oriented Immunology & Inflammation Research, Helmholtz Centre for Infection Research, 38124 Braunschweig, Germany; yanyan.beer@ptb.de; 5Department 3.2 Biochemistry, Physikalisch-Technische Bundesanstalt (PTB), 38116 Braunschweig, Germany

**Keywords:** autophagy, xenophagy, pathogens, pattern recognition receptors, innate immune response

## Abstract

Autophagy is a highly conserved and fundamental cellular process to maintain cellular homeostasis through recycling of defective organelles or proteins. In a response to intracellular pathogens, autophagy further acts as an innate immune response mechanism to eliminate pathogens. This review will discuss recent findings on autophagy as a reaction to intracellular pathogens, such as *Salmonella typhimurium*, *Listeria monocytogenes*, *Mycobacterium tuberculosis*, *Staphylococcus aureus*, and pathogenic *Escherichia coli*. Interestingly, while some of these bacteria have developed methods to use autophagy for their own benefit within the cell, others have developed fascinating mechanisms to evade recognition, to subvert the autophagic pathway, or to escape from autophagy.

## 1. Introduction

The term autophagy is Greek, meaning “eating of myself” and describes diverse processes, such as macroautophagy, microautophagy, chaperone-mediated autophagy, and non-canonical autophagy, in which cells recycle cytoplasmic contents in lysosomes [1]. Macroautophagy, hereafter termed autophagy, is a highly conserved survival mechanism in eukaryotic cells and is mediated by AuTophaGy related (ATG) proteins [2]. During autophagy, cytoplasmic components such as misfolded proteins or damaged organelles are engulfed into double-membrane vesicles, the so-called autophagosomes. Autophagosomes fuse with lysosomes, which leads to the degradation of the sequestered autophagosomal cargo and its recycling for anabolic processes.

In addition to its functional role in cell homeostasis, autophagy defends the host against intracellular pathogens. Furthermore, selective autophagy, mediated by ubiquitination and co-localisation with autophagy receptor proteins, such as sequestosome 1 (SQSTM1), nuclear dot protein 52 (NDP52/ CALCOCO2) and optineurin (OPTN) can act as a direct antimicrobial mechanism, called xenophagy [3].

In 1984, Rikihisa was the first to describe the induction of autophagy by bacteria [4]. Since then, an extensive number of studies demonstrated the role of xenophagy as a defence mechanism and have elucidated how pathogens block the autophagic machinery to escape targeting [5,6], interfere with the fusion of the autophagosome and the lysosome [7,8], or use it to establish replicative niches in different cell types [9,10,11]. In this review, we will discuss the subversion of autophagy by bacterial pathogens.

## 2. Canonical and Selective Autophagy

Autophagy can be subdivided into six key stages. These are (1) signal induction through nutrient deprivation or recognition of specific cargo, e.g., damaged mitochondria or bacteria; (2) phagophore nucleation; (3) cargo targeting; (4) vesicle expansion and autophagosome formation; (5) fusion with the lysosome and finally (6) cargo degradation and nutrient recycling (Figure 1).

### 2.1. Signal Induction

Two key regulators that are involved in the upstream signalling and regulation of autophagy are the mammalian target of rapamycin complex 1 (mTOR) and the AMP-activated kinase (AMPK) as energy sensors. Under nutrient-rich conditions, mTOR phosphorylates ATG13 and Uncoordinated-51-like kinase 1 (ULK1), thus preventing them from interacting with each other [12]. Upon nutrient starvation, mTOR is inactivated. This leads to the activation of ULK1 through dephosphorylation. This, in turn, leads to the translocation of the ULK1 complex comprising ULK1, FAK family kinase-interacting protein of 200 kDa (FIP200), ATG13, and ATG101 to the phagophore assembly site (PAS), followed by autophosphorylation of ULK1, which ultimately results in the phosphorylation of ATG13 and FIP200 [12,13] (Figure 1).

### 2.2. Phagophore Nucleation

In mammalian cells, phagophores are initiated from organelles such as the Golgi complex, the plasma membrane and recycling endosomes [14,15]. The ULK1 complex recruits the class III phosphatidylinositol 3-kinase (PtdIns3K; also known as PI3K) complex consisting of Beclin1, ATG14L1, the vesicular protein sorting 34 (Vps34) and vesicular protein sorting 15 (Vps15) to the PAS, where they are involved in membrane nucleation [15]. The activated PtdIns3-kinase class III (PtdIns3KC3) produces phosphatidylinositol 3-phosphate (PtdIns3P). This recruits effectors such as the PtdIns3P-binding proteins WD-repeat domain phosphoinositide-interacting 1 (WIPI1) and 2 (WIPI2) to facilitate phagophore maturation [15,16] (Figure 1). Interestingly, other regulatory proteins can form a complex with Vps34 and Beclin1 to either promote autophagy, such as UVRAG (UV radiation resistance-associated gene protein), Ambra1 (Activating molecule in Beclin1-regulated autophagy protein 1), and ATG14L [17,18] or to inhibit autophagy, such as Rubicon (Run domain Beclin1-interacting and cysteine-rich domain-containing protein) and Bcl-2 (B cell leukemia/lymphoma 2) [19,20]. However, Rubicon is required for LC3 (microtubule-associated protein light chain 3)-associated phagocytosis (LAP). Several studies showed that Rubicon-deficient cells perform normal phagocytosis but are unable to recruit LC3-II to the cargo-containing phagosome [21,22,23].

### 2.3. Cargo Targeting

Apart from starvation-induced autophagy that engulfs cytoplasmic content non-specifically, certain signals or cytoplasmic events can induce highly selective uptake of distinct cellular structures, such as damaged mitochondria (mitophagy), bacteria (xenophagy), and others. This selective autophagy requires the labelling of cargo with ubiquitin chains. These are then recognised by autophagy receptors, e.g., sequestosome 1 (SQSTM1/p62) that tether the cargo to the autophagic membrane via their LC3-interacting region (LIR) [24,25] and ubiquitin with their ubiquitin-binding domain, respectively [24,25]. In selective autophagy, ULK1 is activated in an mTOR-independent manner, and subsequently, it is recruited to the labelled cargo. However, this process still needs to be characterised in more detail [25] (Figure 2).

### 2.4. Autophagosome Formation

Autophagosome elongation is mediated by two ubiquitin-like conjugation systems, the ATG5-ATG12 conjugation system and the LC3 processing system [14,26]. In the ATG5-ATG12 conjugation system, ATG7 functions as an E1 ubiquitin-activating enzyme activating ATG12 by binding to its carboxyterminal glycine residue in an ATP-dependent manner [27]. Then, the E2-like ubiquitin carrier protein ATG10 covalently attaches ATG12 to the C-terminus of ATG5 to form the ATG12-ATG5 complex. Consequently, the scaffold protein ATG16L1 associates with the N-terminus of ATG5 to form the multimeric ATG12-ATG5-ATG16L1 complex, which affiliates with the expanding phagophore. The conjugation of ATG12 and ATG5 is independent of autophagy induction, and once the phagosome is built, ATG12-ATG5-ATG16L1 dissociates from the membranes [28].

The second ubiquitin-like conjugation system involved in autophagosome formation processes microtubule-associated protein light chain 3 (LC3B). LC3B is present in almost all cell types and has a diffuse cytosolic distribution pattern. Upon autophagy induction, LC3 is cleaved by the cysteine protease ATG4 at its C-terminus to form LC3-I, resulting in a C-terminal glycine residue. This glycine residue is then activated in an ATP-dependent manner by ATG7. The activated LC3-I is then transferred to the E2-like carrier enzyme ATG3 to generate LC3-II after conjugation of phosphatidylethanolamine (PE) to the carboxyl glycine. This lipidated form of LC3 is recruited and integrated into both, the internal and external membrane of the autophagosomes and plays a role in the fusion of membranes as well as in the selection of cargo for degradation (Figure 1). LC3-II is eventually removed by ATG4 before its fusion with late endosomes or lysosomes [1,29]. The changes in synthesis and processing of LC3 makes it an important marker for autophagic flux in cells [28].

### 2.5. Fusion with the Lysosome

Following expansion and closure of the phagophore to build autophagosomes, the autophagosome undergoes maturation to form the autophagolysosome. This includes the removal of ATGs from the outer autophagosomal membrane and the recruitment of machinery responsible for lysosomal delivery that mediates the fusion with the lysosome. In eukaryotes, the fusion of the autophagosome and the lysosome requires the small GTPase Rab7 [30], the soluble N-ethylmaleimide-sensitive-factor attachment receptor (SNARE) protein syntaxin 17 [31] and the SNARE vesicle-associated membrane protein 8 (VAMP8), as well as lysosomal-associated membrane glycoproteins (LAMPs) [32,33]. However, these processes are poorly understood and remain to be further elucidated. Finally, the lysosomal destruction of the autophagosomal membrane and the autophagosomal content is mediated by lysosomal hydrolases.

### 2.6. Alternative Autophagy

Although ATG5 and ATG7 are required for starvation-induced autophagy, an ATG5/ATG7-independent alternative autophagy pathway was recently described [34]. While both canonical and alternative autophagy are under the control of ULK1, there are various differences between these pathways. The main differences are that the only currently known stimulus for alternative autophagy is genotoxic stress, that the phagophore in alternative autophagy derives mainly from the trans-Golgi apparatus and canonical and that alternative autophagy can target diverse substrates under the same stimulus conditions [34,35,36]. Torri et al. recently discovered that upon induction of alternative autophagy, the phosphorylation of ULK1 occurs in a receptor-interacting protein 3 (RIPK3)-dependent manner, leading to the dissociation of ULK1 from the components required for canonical autophagy. ULK1 then moves to the Golgi apparatus to take part in the alternative autophagic pathway [37].

## 3. Autophagy as an Antibacterial Defence

As mentioned above, intracellular bacteria are targets for selective autophagy. They can be targeted either directly in the cytosol, in a process called xenophagy, or in vacuoles and phagosomes, by LC3-associated phagocytosis (LAP). LAP recruits the autophagic machinery while the bacterium is captured in the phagosome (Figure 2). Hence, autophagy can act as an innate immune response mechanism against bacterial infection [38,39,40].

Pattern Recognition Receptor (PRR) signalling is induced following the recognition of damage-associated molecular patterns (DAMPs) and pathogen-associated molecular patterns (PAMPs), leading to the activation of autophagy [41,42]. Alternatively, autophagy is initiated either during the adhesion- and pathogen-induced uptake of bacteria into the host cell, during phagocytosis, or following the damage to the newly formed vacuoles and escape of the bacteria into the cytoplasm [43,44,45] (Figure 3).

### 3.1. Toll-Like Receptors (TLRs)

TLRs are the best characterised PRRs. All known mammalian TLRs are type I integral membrane glycoproteins comprised of an extracellular domain with leucine-rich repeats (LRRs) which mediate ligand recognition and a cytoplasmic Toll/Interleukin-I receptor homology (TIR) domain necessary for signalling initiation [46]. Acting as homo- or heterodimers, they recognise diverse microbial components of bacteria, viruses, fungi, and parasites [46]. The TLRs 1, 2, 4, 5, and 6 are located mostly on the cell surface and predominantly recognise bacterial components. TLRs 3, 7, 8, and 9 are mainly found within the endocytic compartments and recognise viral products [47]. For example, TLR4 homodimerises and recognises lipopolysaccharide (LPS) of Gram-negative bacteria, whereas TLR2 can heterodimerise either with TLR1 or TLR6 to sense bacterial triacylated lipopeptides or diacylated lipopeptides, respectively [48,49].

In general, when TLRs (except TLR3) are activated, they interact with myeloid differentiation primary response 88 (MyD88) via its cytoplasmic domain, which in turn binds IL-1R-associated kinase 4 (IRAK4). IRAK4 then recruits IRAK1, which, after phosphorylation, is released from the complex to bind TNF receptor-associated factor 6 (TRAF6) [50]. This activates nuclear factor (NF) -κB, as well as the mitogen-activated protein kinase (MAPK) kinases (MKKs) [47,51]. Additionally, Beclin1 is ubiquitinated by TRAF6. The ubiquitinated Beclin1 is then released from its inhibitor, B cell lymphoma 2 (Bcl-2), becoming active. Furthermore, TRAF6 activates ULK1 through ubiquitination, and thus induces both autophagy pathways [41,42].

### 3.2. Nucleotide-Binding Oligomerisation Domain-Like Receptors (NLRs)

In addition to TLRs, NLRs are intracellular receptors, that play a pivotal role in the activation of autophagy. Particularly the nucleotide-binding oligomerisation domain proteins (NODs) demonstrate an important function in autophagosome formation. NOD-like proteins recognise bacterial cell wall components, e.g., peptidoglycan, in the cytosol [52,53]. After stimulation of either NOD1 or NOD2, ULK1 is activated by binding the adaptor RIPK2, leading to the recruitment of ATG16L1 to the plasma membrane at the bacterial entry site [54,55]. Furthermore, NOD2 interacts with the immunity-related GTPase family M protein (IRGM), which acts as an activator of AMPK to further activate ULK1 and Beclin1 for autophagosome formation [56].

Additionally, other NLRs can also interact with the autophagic machinery. NLR family CARD domain-containing protein 4 (NLRC4), NLR family pyrin domain containing 3 (NLRP3), NLRP4, and NLRP10 interact with Beclin1, which enables the initiation of Beclin1-mediated autophagic responses [57]. Collectively, NLRs can assemble autophagic factors close to invading microorganisms leading to autophagy activation.

### 3.3. SQSTM1/p62 Like Receptors (SLRs)

As mentioned previously, pathogens can also be identified by the SQSTM1/p62 like receptors (SLRs), also known as autophagy receptors for selective autophagic degradation. The best-known SLRs are sequestosome 1 (SQSTM1/p62), NBR1, optineurin (OPTN) [58], nuclear dot protein 52 (NDP52, also known as CALCOCO2) [59], and NDP52-like receptor TAX1-binding protein 1 (TAX1BP1 or CALCOCO3) [60]. In general, all SLRs consist of two separate domains, the cargo recognition domain, which senses galectin [61] or ubiquitin [62], and the LC3-interacting region (LIR) domain required for autophagosome formation [63].

In addition to SLR-mediated autophagy, the complement component C3 can be used as a tool for mediating xenophagy. C3 coats bacteria and transfers them to the cytosol, where C3 binds directly to ATG16L1 stimulating xenophagy, which restricts bacterial proliferation and eliminates them [64].

### 3.4. Tripartite Motifs (TRIMs)

The family of Tripartite motif (TRIM) proteins possesses a transient function in autophagy and immunity. TRIM16, in collaboration with galectin 3, supports the assembly of autophagic protein complexes such as ULK1 and ATG16L1. It also manipulates the nuclear translocation of the key regulator of lysosomal biogenesis, the transcription factor EB (TFEB) [65]. Nevertheless, the role of TRIMs in selective autophagy remains to be elucidated.

### 3.5. Retinoic Acid-Inducible Gene I (RIG-I) Like Receptors (RLRs)

Unlike PRRs, RLRs, such as retinoic acid-inducible gene I (RIG-I) and melanoma differentiation-associated protein 5 (MDA5) detect pathogen-derived (mostly viral) nucleic acids, which leads to the production of type I interferons (IFNs) and other proinflammatory cytokines. RLRs are negatively regulated by autophagy, preventing prolonged activation of the innate immune system to suppress type I IFN responses [66,67,68]. However, downstream of nucleic acid recognition by RLRs, the cytoplasmic adaptor protein stimulator of IFN genes (STING) protein activates TANK-binding kinase 1 (TBK1) signalling and increases type I IFN production, but it can also induce autophagy [68]. Due to the controversial role of RLRs, further investigations are necessary to understand the crosstalk between RLRs and autophagy.

### 3.6. Cytokine-Mediated Autophagy Activation

Additionally, inflammatory cytokines such as interleukin-1β (IL-1β) [69] and T helper 1 (T_H_1) cell-derived cytokines, such as IFN-γ [70,71,72], as well as tumour necrosis factor (TNF-α) [62] and other IL members, were shown to participate in autophagy induction following bacterial invasion. Generally, cytokine expression is induced after bacterial recognition by PRRs. The crosstalk of PRRs and autophagy shows the high complexity of the immune signalling and reflects the integration of autophagy with several immune regulatory systems. For an advanced understanding of these processes, further studies are of great importance.

## 4. Bacterial Manipulation of Autophagy

Originally, autophagy was discovered as a general, nonselective degradation process in response to starvation [73], but it is also a highly selective process for the degradation of intracellular pathogens [4]. However, many bacterial pathogens evolved mechanisms to avoid or manipulate the autophagic machinery for their own benefit. Here, we will describe some of these mechanisms (Figure 4). The best-studied bacteria regarding autophagy are *Salmonella typhimurium* [74,75], *Mycobacterium tuberculosis* [75,76], *Legionella pneumophila* [75,77,78], *Listeria monocytogenes* [75,79], *Coxiella burnetii* [78], and *Shigella flexneri* [75,80]. Because other reviews have focussed on these bacteria [75,81], we will only provide short summaries of their effects on autophagy here.

### 4.1. Salmonella typhimurium

*S. typhimurium* is a Gram-negative, invasive bacterium, causing foodborne infections. After phagocytic uptake of the bacteria into the host cell, they interfere with the normal biogenesis of the phagolysosome. Most of the bacteria replicate within *Salmonella*-containing vacuoles (SCVs), while some become cytosolic [82]. The cytosolic bacteria are then recognised by the autophagy receptor proteins SQSTM1, NDP52, and OPTN, which target them for degradation through autophagy. Surprisingly, autophagy can also promote the replication of cytosolic *Salmonella* so that the bacteria developed many strategies to activate autophagy [83].

Huett et al. described the E3 ubiquitin ligase LRSAM1, which is crucial for ubiquitin-dependent autophagy of *Salmonella* [84]. Additionally, the production of nitrogen and oxygen species generates 8-nitroguanosine 3′, 5′-cyclic monophosphate (8-nitro-cGMP) which modifies the bacterial surface and drives ubiquitination [85]. Interestingly, *Salmonella* counteracts this ubiquitination and inhibits autophagy. Mesquite et al. characterised the Type III secretion system (T3SS) effector SseL, which inhibits selective autophagy, lowers the autophagic flux and promotes bacterial replication in the cytosol [86]. In a more recent study, Ammanathan et al. showed that the transcription factor EB (TFEB) is suppressed during *Salmonella* infection, as it remains in a phosphorylated state on lysosomes. Additionally, they identified the small molecule acacetin, a flavonoid with anti-peroxidative and anti-inflammatory characteristics. This has a positive effect on xenophagy and acts in an mTOR-independent, but TFEB-dependent manner. Treatment with acacetin results in activated TFEB within the nucleus and TFEB-mediated induction of functional lysosomes targeting SCVs. The resulting increase in proteolytic activity reduces the bacterial burden effectively [87].

### 4.2. Mycobacterium tuberculosis

*M. tuberculosis* (Mtb) is an intracellular bacterium that belongs to the family of Mycobacteriaceae and causes tuberculosis. The bacteria are composed of a waxy membrane that consists of mycolic acids, which makes them resistant to many environmental stresses. Typically, *M. tuberculosis* survives and replicates within macrophages by arresting phagosomal maturation [88]. Another study showed that when recruited to phagosomes, the autophagy machinery restricts the replication of mycobacteria and promotes bacterial degradation [70]. Essential for selective autophagy of mycobacteria are ubiquitination, SQSTM1 and the phosphorylation of SQSTM1 by TBK1 [69], as well as vitamin D [89]. However, the mechanism of how mycobacteria-harbouring phagosomes are recognised by autophagy is not yet fully defined. Watson et al. described ESAT-6, an effector protein secreted via the Type VII secretion system ESX-1, which is critical for pore formation in the phagosomal membrane and autophagy induction [90]. Interestingly, ESAT-6, as well as other effector proteins, can also block autophagic flux [91]. *M. tuberculosis* developed multiple strategies to manipulate autophagy. The Mtb-secreted acid phosphatase SapM targets Rab7, thus preventing autophagosome-lysosome fusion [92]. Another Mtb effector, the enhanced intracellular survival (EIS) N-acetyltransferase, increases the acetylation of the histone H3, leading to upregulation of IL-10 and autophagy inhibition by activation of mTOR [93]. *M. tuberculosis* can be either degraded via autophagy or LAP, which requires Rubicon and the NADPH oxidase 2 instead of ULK1 [94]. *M. tuberculosis* has been shown to be insensitive to both, LAP and the NADPH oxidase. Köster et al. identified that the Mtb effector CpsA causes this autophagic resistance, but the exact mechanism remains to be elucidated [94]. *M. tuberculosis* is able to modulate the expression of diverse host microRNAs (miRNAs), such as miR33, miR125a, and miR17, leading to inhibition of autophagy through the repression of numerous autophagy effectors [95,96,97,98]. Additionally, Mtb induces the expression of miR27a, which targets the ER-located Ca^2+^ transporter CACNA2D3 and inhibits the downstream calcium-associated xenophagy pathway [99]. Kimmey et al. showed, that ATG5 plays an important role in protection against *M. tuberculosis* in infected polymorphonuclear cells (PMNs) and that a loss of ATG5 in PMNs can sensitise mice to *M. tuberculosis*. Interestingly, other ATGs then ATG5 seem not to have a special function in autophagy of *M. tuberculosis* [100].

### 4.3. Legionella pneumophila

*L. pneumophila* is a Gram-negative bacterium that typically colonises aquatic environments but can also infect human lungs, causing Legionnaires’ disease [101]. In infected human macrophages, phagosomes harbouring *Legionella* associate with vesicles from the ER to generate *Legionella*-containing vacuoles (LCVs) that support replication of the bacteria.

The first interaction with autophagy was shown in infected A/J macrophages. There, the LCVs are limited by a double membrane of the ER, co-localise with ATG7 and LC3B and acquire lysosomal markers [102]. However, *Legionella* uses its T4SS to manipulate host autophagy. The bacteria express many effectors that subvert the activity of Rab1, a GTPase that promotes autophagosome biogenesis [103]. Additionally, the cysteine protease RavZ inhibits the maturation of autophagosome-like LCVs via targeting of lipidated ATG8 family members such as LC3-II and their subsequent cleavage. This results in the production of irreversible deconjugated LC3, and thus prevents autophagosome formation [6]. In the past five years, the structure of RavZ and its molecular interaction with LC3 were elucidated in several studies [104,105,106,107]. They showed that the three tandem LIR motifs in the N-terminal region of RavZ form a ß-sheet structure, which provides specific ionic interactions with LC3 [105,107]. Apart from RavZ, *Legionella* effector proteins of the SidE family (SidE, SdeA, SdeB, SdeC) are essential for autophagy inhibition. Bhogaraju et al. and Qiu et al. proposed that the ubiquitin linkage by SidE family members to proteins of the LCV prevents its recognition by autophagy receptors [108,109]. Confirming this theory, a mutant lacking all SidE family members resulted in increased recruitment of SQSTM1 to the LCVs. Surprisingly, even in the absence of RavZ, no increase in LC3B-co-localisation to LCVs was observed, suggesting that other mechanisms are involved in blocking xenophagy that need further elucidation [110].

### 4.4. Listeria monocytogenes

*Listeria* are Gram-positive bacteria causing listeriosis. Typically, listeriosis is a self-limiting foodborne disease, but it can be dangerous for people with immune deficiencies and pregnant women. After invasion, the bacteria escape from the phagosome by expressing the pore-forming toxin listeriolysin O (LLO). The cytosolic bacteria then become motile through the polymerisation of actin tails via expression of the surface protein ActA, which activates the Actin related protein 2/3 (Arp2/3) complex leading to actin reorganisation [111]. Moreover, ActA is important for avoiding autophagy by preventing ubiquitination and recruitment of the autophagy receptor proteins [62]. Interestingly, *Listeria* developed several additional mechanisms to block autophagy. For example, the effectors PlcA and PlcB prevent the targeting of bacteria to autophagy by reducing cellular levels of PtdIns3P, leading to a reduced autophagic flux and bacterial clearance [112]. In addition to the cytosolic replication, *Listeria* also replicates at a reduced rate in macrophage vacuoles, so-called spacious *Listeria*-containing vacuoles (SLAPs). SLAP formation occurs via an LC3-associated phagocytosis pathway and requires LLO activity to damage the membrane of SLAPs and to inhibit fusion with the lysosome [61,113]. Recently, Zhang et al. identified the NOD-like receptor family member NLRX1 as an important receptor for *L. monocytogenes*-induced autophagy. NLRX1is the only NOD-like receptor that contains a mitochondrial target sequence and an additional LC3-interacting region (LIR). *L. monocytogenes* and LLO are both required to induce the oligomerisation of NLRX1, mediating the binding of the LIR to LC3 and hence inducing autophagy [114].

### 4.5. Coxiella burnetii

*Coxiella burnetii* are Gram-negative, obligate intracellular bacteria causing Q fever. In contrast to the pathogens that have developed mechanisms to evade autophagy, *C. burnetii* uses this mechanism for replication within host cells, such as alveolar macrophages. After internalisation, *Coxiella* accumulates LC3B to *Coxiella*-containing vacuoles (CCVs) [115,116]. Surprisingly, the bacteria only activate their metabolism at low pH within the phagolysosome. For LC3B to remain in the CCVs, bacterial replication is required. Thus, *C. burnetii* uses its T4SS to create specialised lysosome-vacuoles to allow bacterial replication [117,118]. Following starvation-induced autophagy in infected macrophages, more infected cells with an increased size and development of the CCVs could be observed [119,120]. Impressively *C. burnetii* uses autophagy for its own benefit to survive in cells. However, the mechanism of its resistance to acid hydrolases is largely unknown. A recent study by Siadous et al. identified the *C. burnetii* vacuolate protein CvpF as a binding partner for Rab26. Rab26 is involved in several processes, such as autophagosome maturation and lysosomal positioning and interacts with ATG16L1. CvpF promotes the accumulation of Rab26 at the CCVs and interacts with it to stimulate and enhance the accumulation of LC3B at the CCVs [121]. Whether ATG16L1 and other co-factors are involved in LC3 lipidation is currently unknown.

### 4.6. Shigella flexneri

*S. flexneri* are Gram-negative, invasive, enteropathogenic bacteria causing strong inflammatory responses and severe watery diarrhoea. Their infection process is similar to that of *Listeria*. After invasion, *S. flexneri* escapes from internalisation vacuoles and polymerises actin for motility [122]. The bacterial surface protein IcsA localises to the cell poles by co-localising with the bacterial actin homologue MreB [123]. This recruits N-WASP, which activates the Arp2/3 complex, resulting in the formation of actin tails [124]. The receptors NOD1 and NOD2, as well as ATG16L1, are then recruited to the bacterial entry site beneath the plasma membrane to initiate autophagy [125]. Additionally, ATG16L1 suppresses the NOD-driven inflammatory responses [55]. *S. flexneri* developed different ways to induce autophagy in infected cells. Apart from NOD activation, ubiquitination, and the recognition by the autophagy receptors SQSTM1, NBR1, and NDP52 are also involved. Independently of the autophagy receptors, ubiquitin induces the vesicle-damage autophagy via the Galectin8-NDP52-LC3 pathway [61]. Another possibility for autophagy activation is the recognition of IcsA by ATG5 leading to binding of the Tectonin domain-containing protein (TECPR1) to ATG5, which promotes autophagosome-lysosome fusion [126]. Interestingly, *S. flexneri* expresses the T3SS effectors IcsB and VirA to avoid autophagy. IcsB binds to IcsA and inhibits the recruitment of ATG5 and TECPR1 [45,62]. In contrast, VirA displays a GTPase-activating protein activity and manipulates Rab2 to inhibit autophagy [5].

### 4.7. Francisella tularensis

*F. tularensis* is a highly virulent intracellular Gram-negative bacterium causing tularemia, also known as rabbit fever. It invades and replicates within various cell types such as macrophages, hepatocytes and pneumocytes [127]. After phagocytosis, *Francisella* escapes from the phagosome and undergoes replication in LC3- and LAMP1-positive *Francisella*-containing vacuoles (FCVs), whose formation is dependent on autophagy [128]. Steele et al. showed that canonical autophagy is induced in mouse embryonic fibroblasts (MEFs) infected with *F. tularensis* due to amino acid starvation. This starvation-induced autophagy degrades proteins to produce amino acids, which are then directly acquired by *F. tularensis* to sustain growth and proliferation. However, *F. tularensis* also efficiently replicates in ATG5-deficient macrophages and mouse embryonic fibroblasts (MEFs). Under this condition, nutrients are generated by an ATG5-independent autophagy pathway, which is not associated with polyubiquitin, LC3, or SQSTM1 [129]. This demonstrates that ATG5-independent autophagy may benefit to some intracellular bacteria by providing nutrients to support bacterial growth (Figure 4A).

### 4.8. Staphylococcus aureus

*S. aureus* are Gram-positive commensal bacteria that are mostly colonising the anterior nares and the skin. However, after damage of epithelial, endothelial, or dermal barriers, it becomes pathogenic and causes a wide range of diseases, such as pneumonia, wound infections, endocarditis, bacteriemia, and sepsis [130]. While originally viewed as an extracellular pathogen, *S. aureus* is also able to invade a variety of cell types, where it can survive and escape phagocytosis (reviewed by Horn et al.) [131]. After the escape from phagocytes, *S. aureus* induces autophagy in host cells, but to date, it is not fully understood whether *S. aureus* subverts or escapes this process. An impressive number of articles was published in recent years to elucidate this phenomenon. For example, several studies showed that *S. aureus*-containing phagosomes are targeted by the autophagic machinery in endothelial cells [132,133]. The pathogen uses this intracellular niche to replicate and escape into the cytosol later in infection to induce host cell death. The *S. aureus* α-toxin was found to be responsible for autophagy induction [133,134]. Maurer et al. showed that an α-toxin-deficient mutant neither recruited LC3 to the bacteria-containing phagosome nor did it escape from the phagosome or prevent lysosome-phagosome fusion [135,136]. The authors hypothesised that the α-toxin perforates the membrane of *S. aureus*-containing phagosomes and autophagy is then induced to protect the cell from the ensuing damage. After treatment with purified α-toxin, ATG5, and calcium-dependent autophagy, but neither PtdIns3K nor Beclin1 were involved, suggesting a non-canonical autophagy pathway [133]. Interestingly, the addition of cAMP to α-toxin-treated or *S. aureus*-infected cells led to a decrease in autophagy [137].

In contrast, Neumann et al. showed the induction of selective autophagy. Directly after phagosomal escape, *S. aureus* becomes ubiquitinated, and the receptor proteins SQSTM1, CALCOCO2 (NDP52), and OPTN are recruited, leading to phagophore formation in murine fibroblasts. However, *S. aureus* can escape from the autophagosome by an unknown mechanism. Additionally, the phosphorylation of the p38 MAPK (MAPK14) was observed, preventing fusion of the lysosome with the autophagosome. After the escape from the autophagosome, cytosolic replication of *S. aureus* was observed [8] (Figure 4B).

A more recent study by Bravo-Santano et al. demonstrated strongly increased AMPK and ERK (extracellular-signal regulated kinase) phosphorylation levels in numerous infected cell types suggesting activation of these kinase pathways. Interestingly, most autophagosomes detected in infected cells did not contain bacteria, suggesting that *S. aureus* induced the autophagic flux for energy generation and nutrient acquisition [138].

### 4.9. Yersinia sp.

*Y. pestis* is a Gram-negative bacterium causing bubonic and pneumonic plague [139]. It has been demonstrated that *Y. pestis*-containing vacuoles (YCVs) in macrophages co-localise with LC3B and fail to reach a pH below pH 7 [140]. The latter finding suggests that *Y. pestis* avoids autophagy by preventing autophagosome-lysosome fusion and thus promotes its own survival in macrophages. Pujol et al. also showed that *Y. pestis* multiplies in bone marrow derived macrophages (BMDMs) with a subpopulation of the bacteria located in double-membrane compartments. In response to *Y. pestis* infection, the conversion of LC3-I to LC3-II was increased, the latter of which was subsequently recruited to the YCVs [141]. Additionally, the bacteria replicate within phagosomes. Interestingly, the replication of *Y. pestis* was not decreased in ATG5-deficient macrophages suggesting that autophagy is not required for bacterial survival in macrophages [141]. Several studies showed that small Rab GTPases are key effectors which help to avoid the killing of *Y. pestis* by macrophages [142,143]. Connor et al. used siRNA-mediated silencing of Rab1b in RAW 264.7 macrophages and showed a reduced survival of *Y. pestis* after infection. Rab1b is directly recruited to the YCVs and blocks the acidification and fusion with LAMP-1 positive structures. However, the bacterial entry and the recruitment of LC3 are not affected, suggesting that Rab1b affects a signalling pathway that is not directly connected to autophagosome formation [142]. In a more recent study, Connor et al. elucidated the functions of Rab4 and Rab11b, which are also recruited to the YCVs and contribute to *Y. pestis* survival. During an early stage of infection, Rab4 cooperates with Rab1b to inhibit the fusion of YCVs with lysosomes. Rab11b has been shown to be sequestered in YCVs, which initiates the disruption of the host cell endosomal pathway via a currently unknown mechanism, in favour of bacterial replication [143].

*Y. pseudotuberculosis* is a Gram-negative bacterium causing enterocolitis, pseudoappendicitis, and mesenteric lymphadenitis. After invasion of macrophages, the bacteria activate autophagy in a T3SS-independent manner. Additionally, it was shown that the bacteria survive and replicate inside autophagosomes. In contrast to *Y. pestis*, the replication rate of *Y. pseudotuberculosis* was decreased in ATG5-deficient MEFs compared to infected wildtype MEFs. This highlights the importance of autophagy for the intracellular growth of *Y. pseudotuberculosis* [144]. Like *Y. pestis*, *Y. pseudotuberculosis* also prevents autophagosome maturation, and thus its own degradation by inhibiting V-ATPase [145] (Figure 4C). However, autophagosomes without bacteria readily fuse with lysosomes indicating that the basal autophagic flux is not impaired [144,145]. Similar to macrophages, HeLa cells infected with *Y. pseudotuberculosis* also activate autophagy. Here, the maturation of YCVs is also inhibited which favours bacterial replication in non-acidic compartments. LC3B is directly recruited to the YCVs to form single-membrane complexes, so-called LAPasomes [146]. Here, vesicle-associated membrane proteins (VAMPs), in particular, VAMP7 play a pivotal role in recruiting LC3B to *Y. pseudotuberculosis*-containing vesicles (YCVs). VAMP3 controls the delivery of bacteria into LC3-positive compartments [146]. The bacterial effectors responsible for inhibition of the autophagosome maturation as well as mechanisms for replication inside YCVs are currently unknown and need to be further elucidated.

*Yersinia enterocolitica* is closely related to *Y. pseudotuberculosis* and causes similar diseases. Comparable to the other *Yersinia* species, it activates autophagy in macrophages and epithelial cells after infection. Deuretzbacher et al. observed intracellular bacteria in double-membrane compartments as well as bacteria associated with LC3-positive vesicles, together with a conversion of LC3-I to LC3-II [147]. The activity of the T3SS, the interference with invasion and ß1-integrin-mediated signalling or the absence of invasion lead to a reduction in bacterial numbers and a decreased activation of autophagy coupled with a reduced number of bacteria in autophagosomes. This suggests that autophagy promotes the killing of *Y. enterocolitica* instead of creation of a replicative niche. In their study, Deuretzbacher et al. showed that the inhibition of autophagy requires the T3SS effector YopE, which targets Rho GTPases for subsequent inactivation [147]. In a more recent study, Valencia-Lopez et al. infected HeLa cells with heat-killed bacteria which were not recruiting LC3 to the YCVs and acidification took part, suggesting that metabolically active bacteria are required for autophagy induction and inhibition of acidification [148]. Additionally, only half of the bacteria are captured in phagosomes, which are LAMP-1-positive, but LC3-negative, indicating that *Y. enterocolitica* is eliminated via the classical lysosomal pathway [148].

### 4.10. Orientia tsutsugamushi

*Orientia tsutsugamushi* are obligate intracellular Gram-negative bacteria which are mainly transmitted from chigger mites to humans. In humans, they cause scrub typhus, the most common febrile illness in the Asia-Pacific region [149]. Patients can develop severe diseases such as pneumonia, meningitis and renal failure and the mortality is moderately high with 30% [150,151]. *O. tsutsugamushi* infects several cell types, such as endothelial cells, fibroblasts, macrophages, polymorphonuclear leukocytes, and dendritic cells [152,153,154]. After invasion of the bacteria into host cells, they rapidly escape from endosomes and replicate within the cytosol [155]. Ko et al. showed that HeLa cells and RAW macrophages infected with *O. tsutsugamushi* display increased autophagy. Interestingly, most bacteria did not co-localise with LC3-II positive autophagosomes, which implicates that autophagy is induced, but the bacteria were not targeted for degradation (Figure 4D). However, after treatment with tetracycline, bacterial evasion from autophagy was suppressed, and *O. tsutsugamushi* was targeted for autophagy and captured in autophagosomes. This suggests that the bacteria express so-far unknown factors that block their recognition by autophagy receptors. Additionally, autophagy does not affect the cellular proliferation of the bacteria [155].

### 4.11. Anaplasma phagocytophilum

*A. phagocytophilum* is a Gram-negative, obligatory intracellular pathogen, causing granulocytic anaplasmosis [156,157,158]. The bacteria infect and replicate inside endothelial cells and granulocytes [156,159]. After endocytosis, they reside and replicate inside membrane-bound inclusions lacking endosomal or lysosomal markers. Additionally, a co-localisation of Beclin1, LC3 and the *A. phagocytophilum*-containing inclusion was observed. Interestingly, while LC3-II increased during infection, no co-localisation of *A. phagocytophilum*-containing inclusions with lysosomal markers occurred, implying that *A. phagocytophilum* subverts autophagy to replicate in an early autophagosome-like compartment separated from lysosomes [160]. *A. phagocytophilum* secretes the Type IV effector protein Ats-1 (*Anaplasma* translocated substrate-1) into the cytoplasm of the host cell [161]. Ats-1 then binds to Beclin1, which induces autophagosome formation. The autophagosomes are recruited to the *Anaplasma*-containing inclusions resulting in the fusion of both forming amphisomes. The autophagic cargo is thereby delivered into the inclusions, providing nutrients to support the proliferation of *A. phagocytophilum* [161,162] (Figure 4E).

### 4.12. Ehrlichia chaffeensis

*E. chaffeensis* are Gram-negative, obligatory intracellular pathogens, which preferentially infect mononuclear phagocytes [163,164]. These pathogens cause the tick-borne zoonosis human monocytotropic ehrlichiosis, a disease with symptoms such as fever, meningitis, thrombocytopenia and, in the worst case, multi-organ failure [165,166]. After infection, the bacteria reside in membrane-bound vacuoles with early endosome-like characteristics, which lack endosomal and lysosomal markers [167,168,169]. To obtain nutrients from the host cell, *E. chaffeensis* uses its T4SS effector Etf-1 (*Ehrlichia* translocated factor-1) to induce Rab5-regulated autophagy. The autophagosome then fuses with the *Ehrlichia*-containing vacuole (ECV) to form amphisomes and provide nutrients for bacterial growth [170]. Lin et al. showed the interaction of Etf-1 with Beclin1, VPS34 and Rab5, which leads to the activation of the class III PtdIns3K and consequently results in autophagy induction [170] (Figure 4F). A more recent study by Lina et al. describes that *E. chaffeensis* uses tandem repeat protein (TRP) effectors to activate Wnt (wingless-related integration site) and PI3K/Akt signalling pathways to modulate downstream signalling and ultimately, to activate mTOR, thereby inhibiting autophagy. Additionally, Wnt signalling promotes the nuclear translocation of TFEB [171]. This displays another mechanism to inhibit autolysosome generation and degradation though canonical autophagy.

### 4.13. Pathogenic E. coli

Uropathogenic *E. coli* (UPEC) cause recurring urinary tract infection (UTI). UPEC invades and persists in bladder epithelial cells. It interacts with ferritin-bound iron to shuttle to the autophagosomal and lysosomal compartments, where it replicates [172]. UPEC was further shown to increase the expression of the small GTPase Rab35, which plays an important role in the endosomal recycling of the transferrin receptor. Interestingly, this receptor is important for the cellular uptake of iron. Both Rab35 and the transferrin receptor are recruited to the UPEC-containing vacuole (UCV), where they increase the levels of iron and, hence, support UPEC survival [173]. ATG16L1 deficiency was shown to protect the host against infection with UPEC in vivo [174]. Conditional knockout mice cleared UPEC infections much more rapidly than wildtype mice and latency was inhibited, suggesting that UPEC uses autophagy to promote its survival and persistence. Clearance of infection was further associated with an increased number of recruited immune cells and a robust inflammatory response and a significant increase in tissue IL-6 and serum IL-1α levels. The ATG16L1 T300A variant found in humans was shown to mimic ATG16L1 deficiency in mice. Expression of this variant in knockout mice abolished UPEC persistence and hence, UTI recurrence [175]. Furthermore, loss of ATG7 but not ATG14 or ectopic P granules protein 5 (EPG5) led to a decrease in intracellular UPEC in bladder epithelial cells [175].

Adherent-invasive *E. coli* (AIEC) is a gastrointestinal pathogen found in excessive amounts in the intestinal tract of Crohn’s disease (CDs) patients. Interestingly, many autophagy-related proteins have been implicated in the disease pathology of CD, a chronic inflammatory disease of the gut. AIEC adhere to and invade into the intestinal epithelial cells [176] and can survive and replicate inside immune cells such as macrophages [177]. Infection of cells with AIEC leads to a significant increase in the expression of proinflammatory cytokines, resulting in intestinal inflammation [177,178].

AIEC replication is restricted by autophagy in both intestinal epithelial cells [179] and macrophages [180]. In ATG16L1- or IRGM-deficient cells, however, they are able to replicate and persist, which results in enhanced inflammation [179]. IRGM forms complexes with NOD2 after activation through PAMPs, leading to the assembly of activated autophagy regulators such as Ulk1 and Beclin1, thus promoting autophagy [56]. Moreover, in macrophages, NOD2 impairment was shown to affect AIEC survival in addition to ATG16L- and IRGM-deficiency [180].

In neutrophil-like cells, AIEC infection was discovered to block autophagy, allowing bacterial survival, and resulting in a marked increase in IL-8 production [7]. Stimulation of autophagy, in contrast, was able to reduce both AIEC survival and the level of IL-8 expression [7].

Nguyen et al. showed that AIEC infection increased the levels of 2 microRNAs, miR30C and miR130A, in intestinal epithelial cells and mouse enterocytes via activation of NF-κB. These microRNAs resulted in a marked reduction of ATG5 and ATG16L1 levels, inhibiting autophagy. This, in turn, increased AIEC persistence and cytokine production. An inverse correlation between the levels of the miRNAs and ATG5/ATG16L1 was confirmed in biopsy samples from CD patients. Furthermore, inhibition of the miRNAs in vitro restored ATG5/16L1 expression and autophagy [181] (Figure 4G).

Recently, AIEC infection has been shown to decrease the number of SUMO-conjugated proteins in intestinal epithelial cells. Ectopic expression of SUMO, in turn, reduced the ability of AIEC to colonise and survive in cells and hence reduced the AIEC-associated proinflammatory response. Moreover, it could be confirmed that autophagy was suppressed upon inhibition of SUMOylation [182].

Enterohaemorrhagic *E. coli*, in contrast to the other pathogens described so far in this review, is a non-invasive gastrointestinal pathogen. To ensure its attachment to the enterocyte surface, it translocates its own receptor, the translocated intimin receptor (Tir) into the host cell via a T3SS. Once inside the host, Tir integrated into the host cell membrane to interact with intimin, a cell surface protein exposed on the bacterial surface. This interaction results in intimate attachment of the bacterium to the surface of the enterocyte and initiation of intracellular signalling cascades [183]. Interestingly, one of these cascades results in the activation of Protein kinase A (PKA) [184]. PKA, in turn, has been shown to phosphorylate ATG13, which releases the latter from the autophagosome, inhibiting autophagy [185]. Recently, Xue et al. discovered that infection of colonic epithelial cells with EHEC blocks autophagic flux. They went on to show that autophagy inhibition by EHEC is a result of Tir-mediated PKA induction which, in turn, inhibited ERK activation and enhanced PI3K/Akt signalling [186] (Figure 4H).

## 5. Conclusions

Over the past 20 years, several studies have revealed the crucial function of autophagy (xenophagy) as a host innate immune mechanism to target and degrade intracellular pathogens. The interaction between autophagy and intracellular bacteria is an alternating process in which either the bacterial replication can be restricted, and the bacteria degraded, or the bacteria escape, subvert, or hijack the host autophagy machinery (Figure 4).

However, plenty of questions remain. How exactly are intracellular bacteria targeted for autophagy? Why are there so many different mechanisms to induce autophagy? Finally, what methods do bacteria use to manipulate host autophagy? 

Hopefully, future research will elucidate the signalling pathways involved in autophagy induction as well as identify novel bacterial effectors, their targets, and their function. A better understanding of these molecular mechanisms of this host–pathogen interaction will facilitate the development of new strategies to combat intracellular pathogens.

## Figures and Tables

**Figure 1 pathogens-10-00110-f001:**
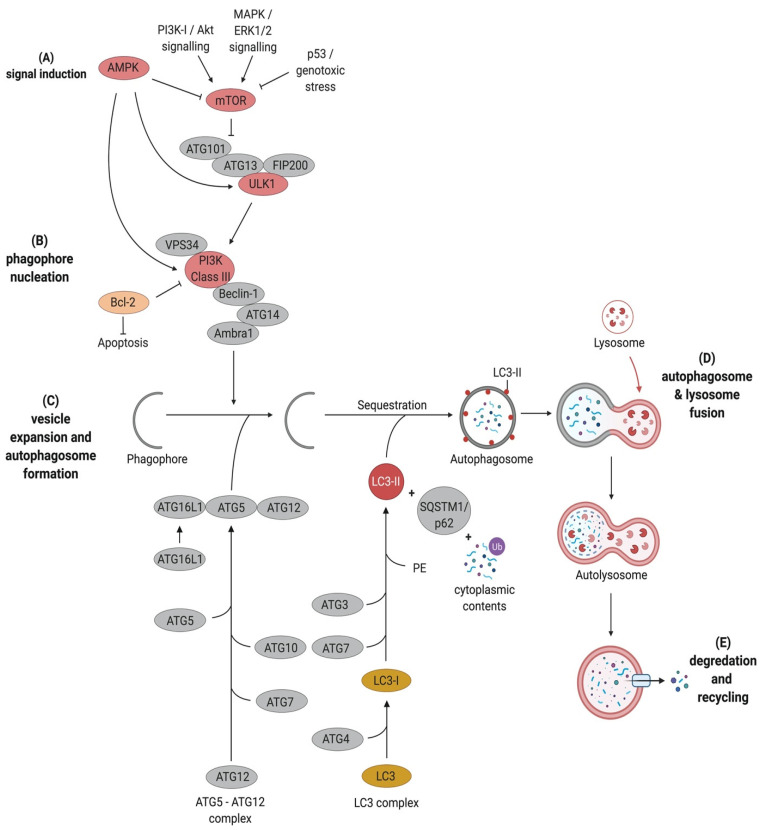
Overview of the autophagy pathway. Autophagy can be subdivided into following stages: (**A**) signal induction, which is regulated by the serine threonine kinases AMPK (AMP-activated kinase) and mTOR (mammalian target of rapamycin complex 1) (**B**) phagophore nucleation, which is induced by the Beclin1-containing phosphatidylinositol 3-kinase (PI3K) class III complex (**C**) vesicle expansion and autophagosome formation, which is governed by two ubiquitin-like conjugation systems, namely, ATG5-ATG12 and LC3 (microtubule-associated protein light chain 3). (**D**) autophagosome and lysosome fusion and (**E**) degradation and recycling. All stages are described in detail in their chapters, respectively (created with biorender.com).

**Figure 2 pathogens-10-00110-f002:**
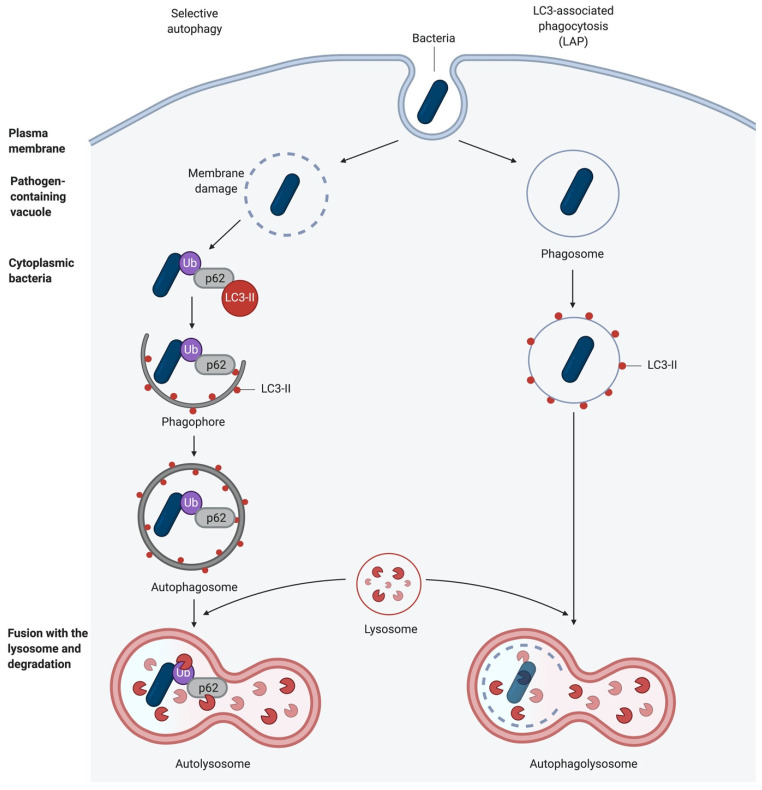
Autophagic elimination of invading bacteria. Cytoplasmic bacteria and bacteria within phagosomes can be degraded through autophagy. After sequestration, the autophagosome fuses with a lysosome to form an autophagolysosome. Some bacteria can escape from the phagosome. These intracellular bacteria are polyubiquitinated and recognised by autophagy receptor proteins to deliver them directly to the phagophores (shown on the left). Bacteria captured in phagosomes are degraded in a process called LC3-associated phagocytosis (LAP; shown on the right) (created with biorender.com).

**Figure 3 pathogens-10-00110-f003:**
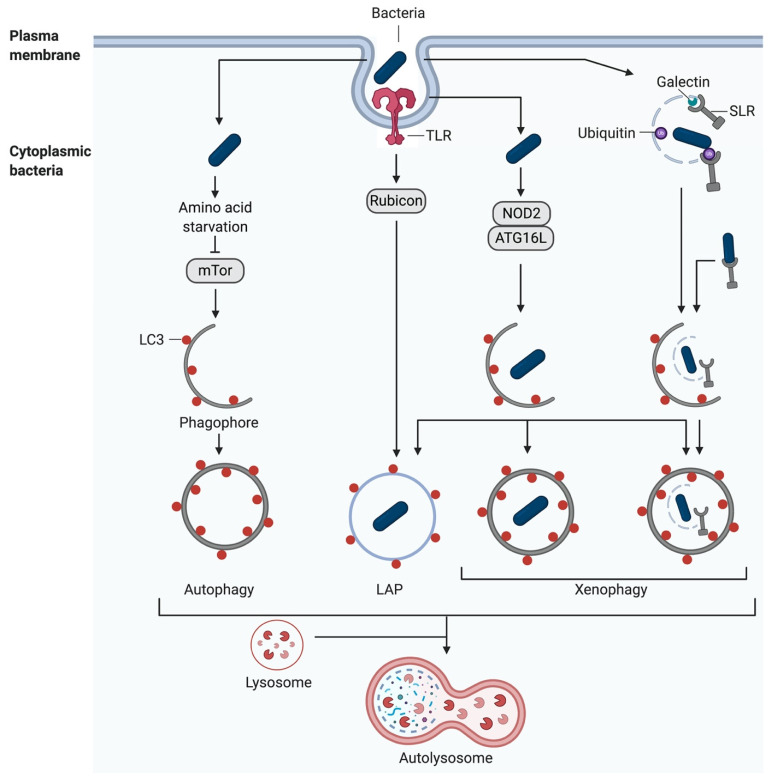
Autophagy as an antibacterial defence mechanism. Autophagy can be induced following the uptake of bacteria into the host cell during phagocytosis or after the escape of the bacteria from the phagosomes into the cytosol. Internalised bacteria can induce xenophagy or LAP by different mechanisms, such as competing for amino acids or by stimulating pattern recognition receptors (PRRs) (created with biorender.com).

**Figure 4 pathogens-10-00110-f004:**
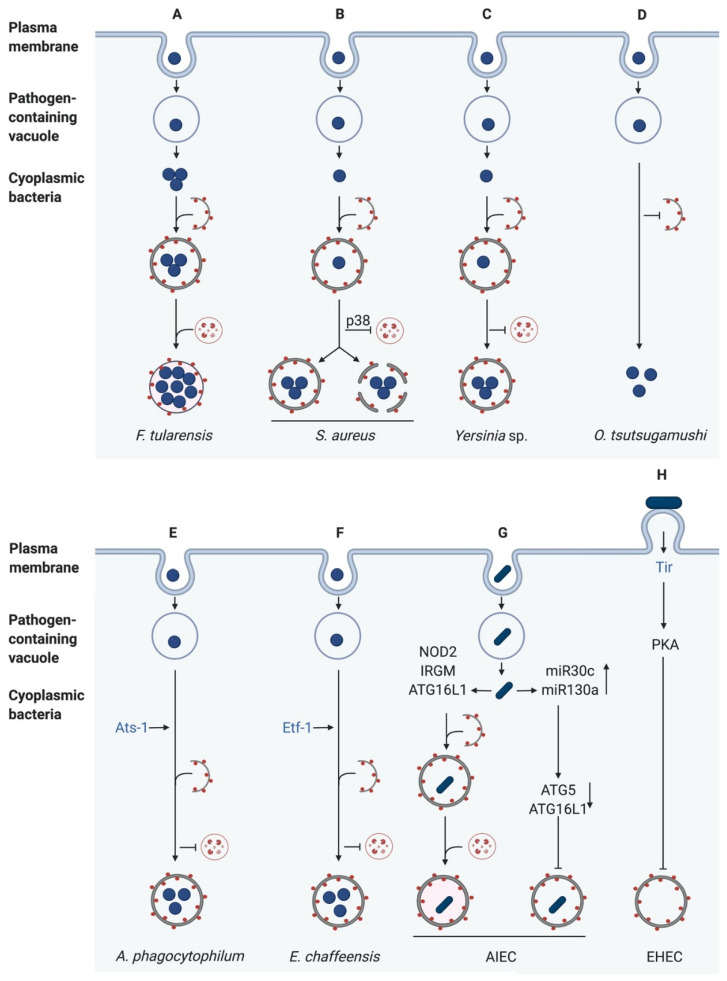
Bacterial evasion or subversion of autophagy pathways. (**A**) *Francisella tularensis* escapes from the phagosome to replicate in the host cytoplasm. Afterwards, it enters the autophagy pathway and replicates in autophagosomes. (**B**) *Staphylococcus aureus* inhibits the autophagosome lysosome fusion through p38 MAPK phosphorylation. Then, it either replicates in autophagosomes or escapes from the autophagosome in an unknown mechanism. (**C**) *Yersinia* sp. block the fusion with lysosomes. Thus, *Yersinia* creates a replicative niche in autophagosomes. (**D**) After infection of host cells, *O. tsutsugamushi* induces autophagy, but the bacteria can prevent their uptake into the autophagosome. (**E**) *Anaplasma phagocytophilum* secretes *Anaplasma* translocated substrate-1 (Ats-1) to induce autophagy. However, the bacteria-containing inclusions fuse with autophagosomes to form amphisomes, delivering nutrients for bacterial replication. (**F**) *Ehrlichia chaffeensis* translocates *Ehrlichia* translocated factor-1 (Etf-1) to induce Rab5-regulated autophagy. This leads to the generation of amphisomes to provide nutrients for bacterial growth. (**G**) While intestinal epithelial cells induce autophagy after invasion of Adherent-invasive *E. coli* (AIEC) due to NOD2 activation, AIEC upregulates host microRNAs to subvert autophagy. (**H**) After Adhesion, Enterohaemorrhagic *E. coli* (EHEC) expresses its Type III secretion system (T3SS), leading to the translocation of translocated intimin receptor (Tir), which activates Protein kinase A (PKA) and thus blocks autophagy formation. LC3-II is displayed as red dots, and bacterial effector proteins are in blue (created with biorender.com).

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
