# Peer review of "Autophagy—A Story of Bacteria Interfering with the Host Cell Degradation Machinery"

_pathogens, 2021, doi:10.3390/pathogens10020110_

Round 1
Reviewer 1 Report
Comments to Authors: In this review, the authors clearly describe the critical processes of autophagy and the relationship of the autophagy machineries with various kinds of infected bacteria. This review will provide enough information to the readers to think about their future studies. The contents of this review are valuable. 1. Characters are extremely small in Figure 1. The reviewer think that it must be hard to recognize for the readers. The authors should redraw characters with a larger size. 2. In Figure 2, characters are extremely small. In addition, there are not enough information to explain each step, authors should explain organelles and each step with relatively large characters. 3. The resolution of Figure 3 is very low. The authors should improve this. 4. Characters in Figure 4 are relatively small and thin, and it is hard to recognize which steps are indicated by each of characters. These will sometime make the readers confuse.

Reviewer 2 Report
The review by Riebisch et al. entitled “Autophagy – A story of bacteria interfering with the host cell degradation machinery” describes how bacterial pathogens manipulate macroautophagy. The review first describes the different cellular steps of autophagosome initiation, development and maturation. In the second part of the review, the authors describe the crosstalk between Pattern Recognition Receptors and autophagy. Finally, the authors list different bacterial pathogens and how these interfere with macroautophagy. Although the scope of the review is interesting, some information would increase the impact of the review such as mechanistic aspects of autophagy manipulation by bacterial pathogens and emerging pathogens not described in the review (see below). This would definitely strengthen the manuscript.
Major corrections
The authors acknowledge that many formerly published reviews describe in-depth the interplay between the described pathogens and autophagic processes. In order to increase the interest of the review, I would recommend not only to provide a “short summary” (p9 lane 287-288) but to implement new articles that enriched the field of autophagy manipulation by bacterial pathogens since the publication of the aforementioned reviews.
Examples: for Legionella: Omotade and Roy, 2020; Salmonella: Ammanathan et al, 2020; Coxiella: Siadous et al, 2020; Shigella: Krokowski and Mostowy, 2019; so on and so forth.
P.10 lane 317-318: ESAT-6 is secreted by the Type VII secretion system ESX-1 and not a T4SS.
Figure 4: as mentioned p.14 lane 515-516, EHEC is a non-invasive pathogen, it is thus surprising and incorrect to observe endocytosed EHEC that ends up in an autophagosome on Figure 4.
Additional intracellular bacterial pathogens manipulating autophagy are missing from the review and dedicated paragraphs should be implemented: Anaplasma phagocytophilum, Orientia tsutsugamushi, Ehrlichia chaffeensis
Minor corrections
Figures 1, 2 and 4 need editing to improve their quality. For example, the font in the figures is too small.
In figure 2, it is unclear where LC3 is located during p62-mediated cargo targeting, does it remain bound to p62 or to the phagophore membrane? Is it LC3-I or LC3-II? Labeling of LC3 should be consistent in the figure. Furthermore, Figure 2 should be placed in the “Cargo targeting” paragraph and not the “Phagophore nucleation” one.
P.3 lane 71: ubiquitin is misspelled
P.3 lane 83: “also” should be removed
Reviewer 3 Report
The review article provides a broad overview of the autophagy degradation pathway and describes interactions between several bacteria and the autophagy pathway.
The title of the article “Autophagy – A story of bacteria interfering with the host cell degradation machinery” is somewhat misleading because as the authors state on line 286 “Because other reviews have focussed [misspelled] on these bacteria [87,88], we will only provide short summaries of their effects on autophagy here.” Indeed, the information included for many of the pathogens reviewed (i.e., Salmonella, Mycobacterium, Legionella, Listeria, and Coxiella) provides little insight into how these bacteria interact or interfere with autophagy.
This article offers a good introduction to host cell sensing of pathogens and initiation of autophagy. Surface-level reporting various bacteria distracts from this information. The article would benefit from greater focus on several pathways and pathogens.
There are numerous primary publications that report the incredible array and complexity of interactions between autophagy and the 10 bacteria selected for this article. The current review article would benefit from more in-depth analysis of details reported in primary publications.
The inclusion of all 10 bacteria within a single review appears overly ambitious. The discussions Salmonella, Mycobacterium, Legionella, Listeria, Coxiella, etc. lack specific information or insight into how bacteria interact with the autophagic pathway. The authors should consider removing the less developed discussions of bacteria, including those for Salmonella, Mycobacterium, Legionella, Listeria, and Coxiella, and bolster the discussion of the remaining bacteria. Alternatively, the authors could rearrange the text according to themes observed in bacteria-autophagy interactions and then use details regarding specific bacteria to exemplify specific themes.
Figures 1, 2 and 4. Text font size is too small and illegible in the printed version. Digital zoom of the digital file further reveals formatting issues for the labels that include irregular spacing between letters in the text. Fig 1. “LC3-II” on the autophagosome is difficult to read when viewed at 300%. All of the text should be enlarged, and a more consistently formatted font should be selected. It appears the figures may have been generated without taking into account size and resolution limitations allowed for articles published by pathogens. Figure 1, 2 and 4 should be remade so that they are interpretable when viewed according to the journals figure size and resolution specifications.
Round 2
Reviewer 2 Report
The authors made the required modifications and improved the quality of the review.
Minor comment: page 21 lane 558, there is an extra "y" in phagocytophilum.